# Optimal Portfolio Liquidation

Yuheng CAI, Taichen FAN
Division of Financial Technology
The Chinese University of Hong Kong
Shatin, Hong Kong
{1155152883, 1155150355}@link.cuhk.edu.hk

December 12, 2020

**Abstract**

Optimal Portfolio Liquidation is one of the common challenges in stock trading whose aim is to minimize a combination of risks and transactions cost arising from permanent and temporary market impact when selling a large number of stocks within a given time frame. Our attempt in this project is to turn this challenge into a reinforcement learning problem and compare the research result with a known baseline method achieved with Almgren and Chriss model[1]. A quick introduction video is here https://youtu.be/dBjzqcKjlBQ

**Keywords:** Almgren and Chriss model   Optimal Liquidation Problem   Deep Reinforcement Learning

## 1   Introduction

Optimal Liquidation Problem - How to Sell Stocks with Minimal Loss. Assume that you have a certain number of stocks or portfolio that you want to sell within a given time frame. Taking into account the costs arising from market impact and a trader's risk aversion, the goal is to create a trading list that sells all stocks in the portfolio, within the given time frame, such that the total cost of trading, also known as expected shortfall is minimized.

## 2   Related Work

Almgren and Chriss[1] provided a direct mathematical solution to the optimal execution of portfolio transactions problem. We follow the same formal mathematical definition for trading trajectory, trading list and trading strategy for liquidating a single stock, price dynamics and market impact etc; but these will be used for the purpose of benchmark. We will explore a reinforcement learning approach that can solve the optimal liquidation problem.

Reinforcement learning is a framework that an intelligent agent explores the financial market in an iterative manner that allows it to learn the optimal trading strategy through new information. Reinforcement learning methods can be categorized into: critic-only, actor-only and actor-critic approach. Our project work will be closely related to the actor-critic with a deep neural network approach[4].

# 3    Preliminaries

Optimal Portfolio Liquidation is essentially aiming at minimizing the combination of risk and transaction costs arising from permanent, temporary market impact and random price fluctuation. As an example, assuming that a financial institute has a large number of stocks that they want to sell within a given time frame. Selling order directly to the market as it is, transaction costs may rise due to permanent and temporary market impact; on the other hand, splitting up into pieces in time, cost may rise due to stock price volatility. Following subsections define how optimal portfolio liquidation problem might be formulated and its mathematical solution based on the Almgren and Chriss Model, which will serve as a baseline method for our Reinforcement Learning approach.

## 3.1    Baseline Method - Almgren and Chriss model

Almgren and Chriss[1] provided a solution to the optimal portfolio liquidation problem by assuming the permanent and temporary market impact functions are linear functions of the rate of trading, and stock prices follow a discrete arithmetic random walk.

## 3.2    Trading Trajectory, Trading List, and Trading Strategy

Suppose we hold $T$ shares of a stock that we need to liquidate before time $T$. Divide $T$ into $N$ intervals of $length = T/N$ and :

- Trading Trajectory: the list $(x_0, ..., x_N)$, where $x_k$ is the number of shares we plan to hold at time $t_k$. It's required that the initial position x0=X, and at liquidation time $T$, $x_N$=0.

- Trading List: the list $(n_0, ..., n_N)$, $n_k = x_{k-1} - x_k$ as the number of shares being sold between time $t_{k-1}$ and $t_k$.

- Trading Strategy: the rule determining $n_k$ from the information available at time $t_{k-1}$.

Following is a visual example of a trading trajectory with $N = 12$

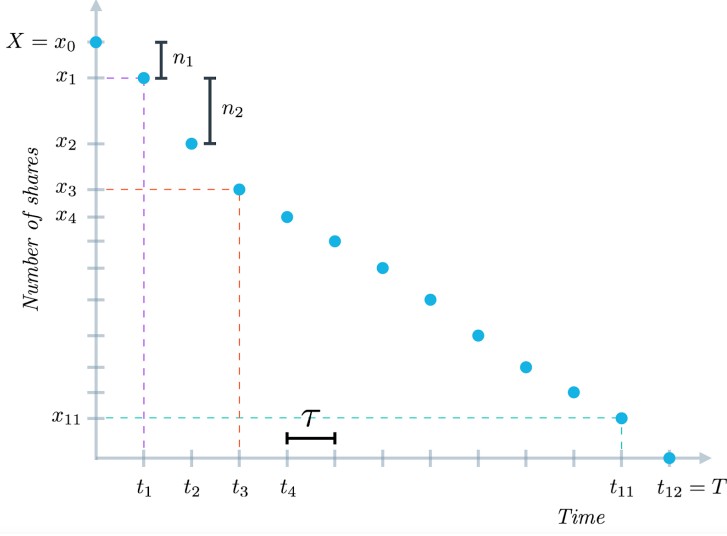

Figure 1: Trading Trajectory Visualization

## 3.3 Price Dynamics

It's assumed that stock price fluctuates according to a discrete arithmetic random walk as following formula:

$$S_k = S_{k-1} + \sigma\tau^{1/2}\epsilon_k$$

for $k = 1, 2, 3, ..., N$ and where:

$S_k$ - stock price at time $k$,

$\sigma$ - standard deviation of the fluctuations in stock price,

$\tau$ - length of discrete time interval,

$\epsilon_k$ - draws from independent random variables.

## 3.4 Permanent Market Impact

Every time we sell a stock, the stock price is affected by market impact. In Almgren and Chris model, it distinguishes between Permanent and Temporary Impact, which will be added into our price model.

Permanent market impact refers to a stock's equilibrium price changes as a result of trading. Its effect persists for the entire liquidation period $T$. We denote the permanent price impact as $g(v)$, and will add it to our price model as following:

$$S_k = S_{k-1} + \sigma\tau^{1/2}\epsilon_k - \tau g(\frac{n_k}{\tau})$$

Where permanent impact function, $g(v)$, is the linear function of the trading

rate, $v = n_k/t$. We will take $g(v)$ to have the form:

$$g(v) = \gamma \frac{n_k}{\tau}$$

where $\gamma$ is a constant, replacing it in above we get:

$$S_k = S_{k-1} + \sigma\tau^{1/2}\epsilon_k - \tau n_k$$

With this form, we can see that for each n shares that we sell, we will depress the stock price permanently by $n\gamma$ regardless of the time we take to sell the stocks.

## 3.5 Temporary Market Impact

Temporary market impact refers to a stock's temporary imbalances in supply and demand caused by our trading. This leads to temporary price movements away from equilibrium. Its effect dissipates by the next trading period. With this, the effective stock price at time k is given by:

$$\tilde{S}_k = S_{k-1} - h\frac{n_k}{\tau}$$

Where, we have again assumed the temporary impact function, $h(v)$, is a linear function of the trading rate, $v = n_k/\tau$. We will take $h(v)$, to have the form:

$$h(v) = \epsilon sign(n_k) + \eta\left(\frac{n_k}{\tau}\right)$$

where $\epsilon$ and $\eta$ are constants. It's important to note that $h(v)$ does not affect the price $S_k$.

## 3.6 Capture

Capture refers to the total profits of trading along a particular trading trajectory till the completion of all trades. Capture can be denoted as:

$$\sum_{k=1}^{N} n_k\tilde{S}_k = XS_0 + \sum_{k=1}^{N}\left(\sigma\tau^{1/2}\epsilon_k - \tau g\left(\frac{n_k}{\tau}\right)\right)x_k - \sum_{k=1}^{N} n_k h\left(\frac{n_k}{\tau}\right)$$

As we can see this is the sum of the product of the number of shares $n_k$ that we sell in each time interval, times the effective price per share $\tilde{S}_k$ received on that sale.

## 3.7 Implementation Shortfall

Implementation Shortfall refers to total cost of trading:

$$I_s = XS_0 - \sum_{k=1}^{N} n_k\tilde{S}_k$$

This is what we need to minimize when determining the best trading strategy.

Since $\epsilon_k$ is a random variable, so is implementation shortfall $I_s$. Therefore minimization of implementation shortfall can be framed in terms of the expectation value of the shortfall and its corresponding variance. We will denote the expected shortfall as $E(x)$ and variance of shortfall as $V(x)$:

$$E(x) = \sum_{k=1}^{N} \tau x_k g(\frac{n_k}{\tau}) + \sum_{k=1}^{N} n_k h(\frac{n_k}{\tau})$$

and

$$V(x) = \sigma^2 \sum_{k=1}^{N} \tau x_k{}^2$$

So now, we can reframe our minimization problem in terms of $E(x)$ and $V(x)$. For a given level of variance of shortfall, $V(x)$, we seek to minimize the expectation of shortfall, $E(x)$.

## 3.8   Utility Function

The goal of optimal portfolio liquidation is to find the strategy that has the minimum expected shortfall $E(x)$ for a given maximum level of variance $V(x) \geq 0$. This constrained optimization problem can be solved by introducing a Lagrange multiplier. Therefore the problem finally reduces to finding the trading strategy that minimize the Utility Function $U(x)$:

$$U(x) = E(x) + \lambda V(x)$$

The parameter $\lambda$ is referred to as trader's risk aversion, which controls how much we penalize the variance relative to the expected shortfall.

The intuition of this utility function can be thought of as follows. Consider a stock, which exhibits high price volatility and thus a high risk of price movement away from the equilibrium price. A risk averse trader would prefer to trade a large portion of the volume immediately, causing a known price impact, rather than risk trading in small increments at successively adverse prices. Alternatively, if the price is expected to be stable over the liquidation period, the trader would rather split the trade into smaller sizes to avoid price impact. The trade-off between speed of execution and risk of price movement is ultimately what governs the structure of the resulting trade list.

Almgren and Chriss solved the above problem and showed that each value of risk aversion there is uniquely determined optimal execution strategy, which is summarized as follows just for completeness. The details can be found in their paper [1].

The optimal trading trajectory:

$$x_j = \frac{sinh(\kappa(T - t_j))}{sinh(\kappa T)} X, \quad for j = 1, 2, 3, ..., N$$

and the associated trading list:

$$n_j = \frac{2sinh(\frac{1}{2}\kappa\tau)}{sinh(\kappa T)} cosh(\kappa(T - t_{j-\frac{1}{2}})) X, \quad for j = 1, 2, 3, ..., N$$

The expected shortfall and variance of the optimal trading strategy are given by:

$$E(x) = \frac{1}{2}\gamma X^2 + \epsilon X + \tilde{\eta} X^2 \frac{tanh(\frac{1}{2}\kappa T)(\tau sinh(2\kappa T) + 2T sinh(\kappa \tau))}{2\tau^2 sinh^2(\kappa T)}$$

$$V(x) = \frac{1}{2}\sigma^2 X^2 \frac{\tau sinh(\kappa T)cosh(\kappa(T-\tau)) - T sinh(\kappa \tau)}{sinh^2(\kappa T)sinh(\kappa \tau)}$$

In this project, we will apply the reinforcement learning approach to find the optimal execution strategy instead of above Almgren and Chriss's mathematical solution, which only serves as a benchmark for our project.

# 4 Reinforcement Learning Approach

## 4.1 Actor-Critic Method

Actor-Critic method is adopted to combine the advantages of actor-only and critic-only methods. In this method, the critic learns the value function and uses it to determine how the actor's policy parameters should be changed, while the critic supplies the actor with knowledge of the performance. Actor-critic methods usually have good convergence properties, in contrast to critic-only methods. The Deep Deterministic Policy Gradients (DDPG) algorithm[2] is one example of actor-critic method that has been explored in this project. The rest of the section formulates the optimal liquidation problem so that it can be solved using reinforcement learning.

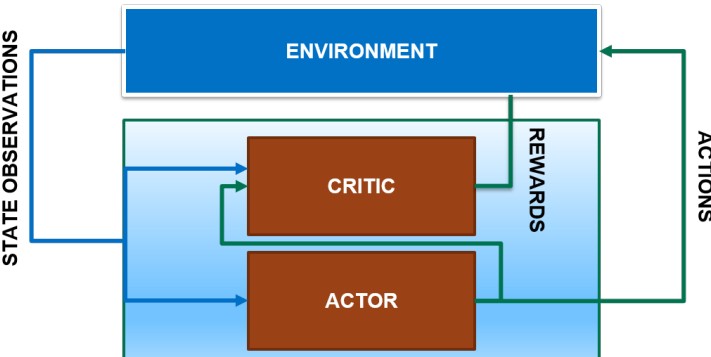

Figure 2: Actor-Critic Method

The optimal liquidation problem is a minimization problem where we need to find the trading list that minimizes the implementation shortfall. In order to solve this problem through reinforcement learning, we need to restate the optimal liquidation problem in terms of States, Actions and Rewards.

## 4.2 States

The optimal liquidation execution problem requires that the agent must sell all stocks in hand within a given time frame. Hence the state vector must provide information about remaining time or the number of trades remaining at any time step. In this project state vector at time $t_k$ as:

$$[r_{k-5}, r_{k-4}, r_{k-3}, r_{k-2}, r_{k-1}, r_k, m_k, i_k]$$

where:

- $r_k = log(\frac{S_k}{S_{k-1}})$ is the log of return at time $t_k$

- $m_k = \frac{N_k}{N}$ is the number of trades remaining at time $t_k$ normalized by the total number of trades

- $i_k = \frac{X_k}{X}$ is the remaining number of shares at time $t_k$ normalized by the total number of shares

This is a simplified state vector capturing minimum variables, which can be further expanded for agents to detect possible price trends and transaction costs etc.

## 4.3 Actions

To generalize the design, we will define action $a_k$ at time $t_k$ as the percentage of total number of stocks to sell. So the exact number of shares to sell at each time step is:

$$n_k = a_k \times x_k$$

where $x_k$ is the number of shares remaining at time $t_k$.

## 4.4 Rewards

As mentioned previously, utility function is defined as:

$$U(x) = E(x) + \lambda V(x)$$

where $E(x)$ the expected shortfall, $V(x)$ is referred as level of variance of shortfall and $\lambda$ refers to trader's risk aversion, which controls how much we penalize the variance relative to the expected shortfall.

Defining the rewards is trickier than defining states and actions, since the original problem is a minimization problem. One option is to use the difference between two consecutive utility functions. By maximizing the difference between two consecutive utility functions ($t$ and $t + 1$), we are effectively driving utility function down over time. So denoting the optimal trading strategy trajectory computed at time $t$ as $x_t^*$, reward at time $t$ as:

$$R_t = \frac{U_t(x_t^*) - U_{t+1}(x_{t+1}^*)}{U_t(x_t^*)}$$

Where we have normalized the difference to train the actor-critic model easier.

## 4.5 Simulation Environment

In order to train our deep reinforcement learning agent implemented in DDPG, we need an environment to simulate stock prices that follow a discrete arithmetic walk and that the permanent and temporary market impact functions are linear functions of the rate of trading, just like in the Almgren and Chriss model. This simple trading environment serves as a starting point to create more complex trading environments such as book orders, network latencies, trading fees etc (for future improvement but not in the scope of this project).

# 5 Experiments

## 5.1 Benchmark Environment Parameters

The simulation environment can be characterized by Financial Parameters and Almgren and Chriss Model Parameters as follows.

| Financial Parameters | | | |
|---|---|---|---|
| **Annual Volatility:** | 12% | **Bid-Ask Spread:** | 0.125 |
| **Daily Volatility:** | 0.8% | **Daily Trading Volume:** | 5,000,000 |

Table 1: Financial Parameters for Benchmarck

Same Financial Parameters are being applied throughout all experiments.

| Almgren and Chriss Model Parameters | | | |
|---|---|---|---|
| Total Number of Shares to Sell: | 1,000,000 | Fixed Cost of Selling per Share: | $0.062 |
| Starting Price per Share: | $50.00 | Trader's Risk Aversion: | 1e-06 |
| Price Impact for Each 1% of Daily Volume Traded: | $2.5e-06 | Permanent Impact Constant: | 2.5e-07 |
| Number of Days to Sell All the Shares: | 60 | Single Step Variance: | 0.144 |
| Number of Trades: | 60 | Time Interval between trades: | 1.0 |

Table 2: Almgren and Chriss Model Parameters for Benchmark

Almgren and Chriss solved the minimization problem mathematically to determine the unique optimal execution strategy for each value of risk aversion $\lambda$.

## 5.2 DDPG V.S. Almgren and Chriss Methods

Deep Deterministic Policy Gradient (DDPG) is implemented in this project to find a policy that can generate optimal trading trajectories minimizing

implementation shortfall and benchmarked against the Almgren and Chriss Model using the same simulation environment. A DDPG agent is fed the states observed from the simulation environment and predicts actions using the actor model and performs trading in the environment. Then the simulation environment returns the reward and new state The process continues for a given number of episodes. Our DDPG implementation has been trained for 100,000 episodes. The comparison between DDPG and Almgren and Chriss Model benchmark is as follows:

| Method | Risk Aversion $\lambda$ | Implementation Shortfall |
|---|---|---|
| Almgren Chriss Model | 1e-06 | $480,925.46 |
| DDPG Model | 1e-06 | $707,143.50 |

Table 3: DDPG V.S. Almgren and Chriss Implementation Shortfall

# 6    Conclusion

In this project we conducted basic experiments to seek optimal portfolio liquidation strategy using Deep Deterministic Policy Gradient method under a simulation environment where stock prices follow a discrete arithmetic walk and that the permanent and temporary market impact functions are linear functions of the rate of trading. Under the same financial and Almgren and Chriss parameters, DDPG achieved a higher (worse) expected implementation shortfall compared with that produced by Almgren and Chriss model.

Although the preliminary result is not satisfactory, there are multiple areas worth exploring in future:

- Incorporate different reward function design to formulate the reinforcement learning problem

- Use more realistic price dynamics, such as geometric brownian motion (GBM) [3]

- Add more complex dynamics to the environment such as trading fees by adding and extra term to the fixed cost of selling

- Try other actor-critic Deep Reinforcement Learning method for continuous space

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
