# OpenReview forum: "Optimal Portfolio Liquidation"
_CUHK.edu.hk/2021/Course/IERG5350_

### Official Review · AnonReviewer3 · 2020-12-15
**This article uses DDPG to solve the problem of Optimal Portfolio Liquidation. Although it is a direct use of existing work, it is qualified as a course project.**

**Rating:** 7
**Confidence:** 4

**Review:**

General:

1. Significance: This article uses DDPG to solve the problem of Optimal Portfolio Liquidation. Although it is a direct use of existing work, it is qualified as a course project.
2. Novelty: Use mature work directly, without any own work and innovation.
3. Technical quality: This paper applies the DDPG method to Optimal Portfolio Liquidation, rationally abstracts and defines the problem, solves certain practical problems, and realizes the reasonable application of the DRL method.
4. Clarity: The structure of this paper is clear, but it lacks a flowchart that reflects the overall method. The analysis of the results is not detailed.

Specific:

1. Pros:
a. Define Optimal Portfolio Liquidation problem.
b. Use DDPG to solve the problem of Optimal Portfolio Liquidation.
2. Cons:
a. The method in this article needs to be further improved to achieve performance beyond the baseline.
b. Use existing work directly, without own work and improvement.
c. There is no mention of the existing project in the related work part, and it is dishonest to directly use other people’s code and projects without quoting and explaining.

---

### Official Review · AnonReviewer1 · 2020-12-18
**As a course project, its workload is enough, but the proposed method is not new and not good enough.**

**Rating:** 6
**Confidence:** 4

**Review:**

General:

Significance: This paper uses Deep Deterministic Policy Gradient (DDPG) to solve the Optimal Liquidation Problem, and compares it with the method proposed by Almgren and Chriss, although the preliminary result shows that the proposed DDPG model cannot beat Almgren Chriss model.

Novelty: Use DDPG to solve the optimal liquidation problem is not innovative enough, since we can find many GitHub repo on this topic. But as a course project, its workload is enough.

Technical quality:  This paper use DDPG to solve the focused problem and conduct the experiments in a simulation environment.

Clarity: The clarity is good and easy to follow, although there are many symbols and formulas. This paper defines the problem clearly and uses pictures to help readers better understand it.

Specific:

Pros:
a. define the Optimal Liquidation Problem clearly;
b. use DDPG to solve the problem;
c. the mentioned future works are worthy to have a try.

Cons:
a. directly use DDPG to solve this problem is not new, maybe you can check this paper: https://arxiv.org/pdf/1906.11046.pdf , which use DDPG-based multi-agent to train the model;
b. maybe you should compare with some state-of-the-art methods since Almgren Chriss model is proposed in 2011.

---

### Official Review · AnonReviewer2 · 2020-12-19
**This paper combines reinforcement learning with real problems, but it should focus more on how they improve existing model rather than just introduce it.**

**Rating:** 6
**Confidence:** 4

**Review:**

Probs:

1.This paper has clear structure and is easy to understand.

2. The authors apply what they have learned in this course to solve real problem.

Cons:

1. In related work, they didn't focus on the previous work, point out the unsolved problems in this field and how their work solve these problems.

2.The paragraph are not aligned and there is no sequence number following the formulas.

3.There is no implementing details. The experimental results and analysis are not sufficient.



Question and suggestion:

1. Write more about the experiment and results.

2. This paper uses other's model. It would be better if they improve the performance of this model.